# Impact of Modifiable Factors Associated with Physical Frailty and Cognitive Impairment Trajectory of Older Adults: Using the Korean Longitudinal Study of Aging 2006–2018

**DOI:** 10.3390/healthcare13030315

**Published:** 2025-02-04

**Authors:** Sumi Lee

**Affiliations:** Department of Nursing Science, Howon University, Gunsan 54058, Republic of Korea; sumilee@howon.ac.kr; Tel.: +82-63-450-7756

**Keywords:** aging, cognitive impairment, frailty, longitudinal study, overweight, obesity, medication, depression

## Abstract

**Background/Objectives**: To identify joint trajectories of physical frailty and cognitive impairment among community-dwelling older adults and to determine modifiable factors for each trajectory. **Methods**: Data were utilized from the Korean Longitudinal Study of Aging, which was conducted between 2006 and 2018. Physical frailty was assessed using the Fried phenotype, and cognitive impairment was evaluated using the Korean version of the Mini-Mental State Examination. Group-based trajectory modeling and logistic regression were employed for the analyses. **Results**: Based on longitudinal data, 415 participants averaging 72.2 years of age were analyzed. Three trajectories of physical frailty were identified: mild physical frailty, moderate physical frailty, and improving frailty. Two trajectories of cognitive impairment were identified: stable cognitive impairment and improving cognitive impairment. Factors influencing physical frailty trajectories included the number of medications taken, being overweight or obese, and depression. Education level was found to be associated with cognitive impairment trajectories. **Conclusions**: This study provides evidence for the distinct identification of joint trajectories of physical frailty and cognitive impairment, which can inform the target groups for intervention. It offers a basis for including modifiable physical and mental factors in intervention components for physical frailty trajectories.

## 1. Introduction

As the older adult population increases, frailty is receiving attention for the health and successful aging of the elderly. According to the results of the Korean Older People Survey, frail older adults accounted for 8.4% and pre-frail older adults accounted for 49.1% [1]. These figures align with global trends, where the global prevalence of frailty among older adults ranges from 4% to 59% [2].

Frailty refers to a state of physiological vulnerability associated with aging, characterized by the accumulation of tissue damage that leads to functional decline [3,4]. While the Fried phenotype [4] remains the gold standard for assessing physical frailty, tools such as the Frailty Index (FI), Tilburg Frailty Indicator (TFI), and Multidimensional Prognostic Index (MPI) [5,6,7] measure frailty from a multidimensional perspective. Identifying factors contributing to frailty and developing effective management strategies are crucial for older adults. Frailty increases the risk of health problems such as reduced ability to perform daily activities, fractures, depression, loneliness, institutionalization, and hospitalization [8,9].

The International Consensus Group has categorized frailty into subtypes, including cognitive frailty, social frailty, oral frailty, and nutritional frailty. This classification highlights the importance of targeted healthcare management in these areas. Particularly, cognitive frailty refers to the coexistence of physical frailty and cognitive impairment, indicating mild cognitive impairment or subjective cognitive decline, which precedes Alzheimer’s disease or other dementias [10]. Since cognitive frailty leads to negative outcomes, such as dementia [11], disability [12], reduced quality of life [13], and even mortality [14], it is essential to focus on prevention and healthy aging. Healthcare providers play a critical role in identifying older adults with cognitive frailty as high-risk groups and providing timely intervention programs to improve health outcomes.

Studies from Spain show that older age and lower educational attainment increase the risk of cognitive frailty [13]. In India, being identified as women, unmarried, and unemployed are risk factors [15]. On the other hand, community engagement was suggested as a protective factor [16]. In Korea, significant correlations were observed between polypharmacy and cognitive frailty [17], as well as increased risk among older adults with heart failure, prolonged sleep duration (>8 h), and depressive symptoms [18]. Previous studies have primarily focused on cross-sectional surveys to explore the incidence rate or determinants of cognitive frailty, which are based on short-term analyses that focus on differences in individual averages [15,19]. This study selected only subjects with cognitive frailty at baseline, focusing specifically on the progression and dynamics within this high-risk group. This approach addresses a gap identified in previous studies [20,21], which struggled to comprehensively capture dynamic changes and processes over time within broader populations.

However, previous studies lack comprehensive research on the general health status of older adults, including physical, psychological, and social factors, which can predict cognitive frailty, as well as adverse outcomes. Aligned with the integral conceptual model of frailty [22], this research emphasizes modifiable factors rather than immutable characteristics such as age, sex, and education level [15,19]. By targeting these modifiable factors, interventions can more effectively prevent and delay cognitive frailty in older adults. To overcome the limitations, health trajectory research has gained attention [23]. Health trajectories involve using longitudinal data collected from individuals or groups at multiple time points to understand the dynamic changes in health conditions over time. Such research supports human-centered care and informed policy making. Group-based trajectory models are methods to identify relationships between various trajectories created by outcome variables [24].

Therefore, this study aims to examine the process of change in cognitive frailty over time using data from the Korean Longitudinal Study of Aging and focusing on modifiable factors. By applying group-based trajectory modeling, the study will explore joint trajectories of physical frailty and cognitive impairment in older adults. Multinomial and binomial logistic regression analyses will be employed for factors influencing these trajectories, providing a theoretical basis for early detection and intervention programs for cognitive frailty.

## 2. Materials and Methods

### 2.1. Study Design

This secondary data analysis study uses data from the Korean Longitudinal Study of Aging (KLOSA) to derive joint trajectories of physical frailty and cognitive impairment in older adults and to identify modifiable factors affecting these trajectories.

### 2.2. Study Participants

This study utilized data from the KLOSA, which was provided from the first wave (2006) to the seventh wave (2018). The KLOSA comprises longitudinal research data collected to understand the actual conditions of older adults in the process of transitioning to an ultra-aged society and to establish effective national policies and interdisciplinary academic research [25]. The KLOSA conducts basic surveys with the same survey items every even year, and completed the seventh basic survey in 2018. The target population of the KLOSA is middle-aged and older adults aged 45 years or older, excluding Jeju Island in 2006, and comprehensively evaluates older adults, including physical examinations, health status, and social functions.

The KLOSA employs a stratified sampling method based on region and housing type, categorizing areas into metropolitan/provincial and urban/rural classifications. The sample survey districts were divided according to the population size and allocated accordingly. This longitudinal dataset is suitable for examining the changes in cognitive frailty among older adults from 2006 to 2018.

### 2.3. Data Collection

The specific selection criteria for the subjects of this study are as follows:(1)Those who were 65 years of age or older at the time of the first-wave survey;(2)Those who responded to the survey and measurements from the first to the seventh waves (with less than 10% missing data [26]);(3)Those who met the criteria for cognitive frailty at the time of the first-wave survey;(4)Those who had not been diagnosed with dementia;(5)Those who participated in the follow-up survey from the second wave to the seventh wave (those who responded to the survey and measurements at least three times out of six follow-up surveys).

The participants for analysis were selected from the collected data according to the selection criteria (Figure 1). The number of participants who participated in the first-wave KLOSA was 10,254. The study excluded 4164 participants who were 65 years or older and 6090 participants who were younger than 65. It also excluded those who did not respond or refused to participate in the survey or measurement of the variables at baseline (2281 participants), those who did not meet the criteria for cognitive frailty in the first wave (1415 participants), and those who did not participate in the follow-up survey from the second wave to the seventh wave (53 participants). As a result, 415 participants were selected for final analysis.

### 2.4. Study Instruments

The tools or questions used to measure the variables included in this study are as follows.

#### 2.4.1. Physical Frailty and Cognitive Impairment

Cognitive frailty is defined when older adults meet the criteria for both physical frailty and cognitive impairment, excluding concurrent dementia or other dementias [10]. Physical frailty was selected for the gold standard Fried phenotype [4], but this study used three of the five indices based on variables available in the KLOSA based on previous research [27]. The study measured physical frailty using three key indicators: reduced handgrip strength, severe fatigue, and unintentional weight loss [27]. Reduced handgrip strength was calculated as the average of two measurements of handgrip strength in the right and left hands. Reduced handgrip strength was classified as less than 28 kg for men and less than 18 kg for women, according to the Asian Working Group of Sarcopenia criteria in 2019 [28]. Severe fatigue was assessed using two questions from the Center for Epidemiological Studies Depression Scale (CES-D) [29]: “Did you find everything you did during the past week difficult?” and “Could you not manage to do anything?” Each question was scored on a 4-point scale, ranging from 1 point for “sometimes (less than a day)” to 4 points for “always (about 5 to 7 days).” Participants with a total score of 6 or higher were classified as having severe fatigue [4]. Unintentional weight loss was measured as a self-reported loss of 5kg or more in the past year [4].

Cognitive impairment was measured using the Mini-Mental State Examination in Korean (MMSE-K), which consists of 19 questions that assess various cognitive domains.

The score range is from 0 to 30 [25]. The older adults with MMSE-K scores of 24 or higher are considered to have no cognitive impairment (normal), those with scores of 18 to 23 are considered to have mild cognitive impairment (cognitive decline), and those with scores of 17 or lower are considered to have severe cognitive impairment (suspected dementia) [30]. In this study, individuals with MMSE-K scores below 24 were considered to have cognitive impairment [31,32].

#### 2.4.2. Demographic Factors

Demographic factors of the participants include age, sex, education level, and marital status, which were measured at baseline. Age was categorized into 65 to 74 years and 75 years or older. Education level was categorized into elementary school graduate or middle school graduate or higher. Marital status was categorized into married or unmarried (separated, divorced, widowed, or single).

#### 2.4.3. Physical Factors

Comorbidities were assessed using survey results data on hypertension, diabetes, chronic lung disease, heart disease, liver disease, cerebrovascular disease, cancer, arthritis and rheumatism, psychiatric disorders, prostate disease, urinary incontinence, and eye diseases. Each disease was coded as 1 if diagnosed by a doctor and 0 otherwise, with the total score ranging from 0 to 12. Polypharmacy was defined as taking three or more medications prescribed by a doctor [30], categorized into 0, 1–2, and 3 or more [33]. Body mass index (BMI) was calculated using measured weight and height and categorized as underweight (<18.5), normal (18.5–23), overweight (23–25), and obese (≥25), according to the WHO Western Pacific Region’s Asia Pacific BMI criteria [34]. Vision and hearing impairment were assessed using self-reported responses to the question on a 5-point scale, with a score of 4 or higher indicating impairment [35]. Physical activity was defined as engaging in exercise at least once a week, and alcohol consumption was defined as current alcohol use.

#### 2.4.4. Psychological Factors

Subjective health status was evaluated using a self-reported question on a five-point scale, with responses ranging from “very good” to “very poor”. Respondents rating their health as “very good” to “average” were considered to have good health, while those rating it as “poor” or “very poor” were considered to have poor health. Depression was measured using eight questions from the Korean version of the Center for Epidemiological Studies Depression Scale (CES-D) [29], excluding two questions overlapping with the definition of physical frailty. Answers were scored on a 4-point scale, with a total score range of 8 to 32, indicating higher levels of depression. As there is no consensus among experts on cutoff points for depression, continuous variables were used to assess levels of depression.

#### 2.4.5. Social Factors

Social engagements refer to the sum of seven social activities at baseline, including current employment status (wage workers, self-employed, and unpaid family workers), religious organizations, social gatherings (senior centers and mutual aid groups), leisure, cultural, and sports groups (senior colleges), volunteering, alumni associations, local community associations, political parties, civic groups, and interest groups. Social engagement means when there is more than one type of social activity.

### 2.5. Statistical Analysis

The study used statistical software packages SPSS 26.0 (Windows, Armonk, NY, USA) and STATA 17.0 (Stata Corp., College Station, TX, USA). To understand the general characteristics of the participants, percentage, mean, *t*-test, and chi-square tests were conducted. The number of trajectory groups with similar patterns of physical frailty and cognitive impairment in the group-based trajectory model was determined using goodness-of-fit indices, probabilities of belonging to each subgroup, and meaningfulness of interpretation. Absolute goodness-of-fit indices include Bayesian Information Criteria (BIC), Sample-size adjusted BIC (Sa BIC), and Akaike Information Criterion (AIC). Among them, BIC is a goodness-of-fit index used to compare different models containing various shapes of trajectories or numbers of trajectories obtained from each evaluated model [36]. Smaller absolute values of BIC and AIC indicate simpler and more explanatory models. When the number of groups increases, BIC values also increase. The goodness of fit of the model was evaluated based on the point where the increase in BIC slows down (elbow plot of indicator) [37,38]. Entropy, which represents the quality of classification, is a measure of the accuracy of group classification in the entire sample. It indicates how much classification is problematic. Entropy is based on the posterior probability of classifying each individual into a certain trajectory group and is expressed as a value ranging from 0 to 1. When each individual is accurately classified, it approaches 1, and vice versa [39]. In other words, the closer the value is to 1, the fewer errors there are in group classification [37]. Various functions (e.g., linear, fourth-order functions) and trajectories were also verified. A trajectory model for physical frailty and cognitive impairment that meets the selection criteria was identified, and a joint trajectory of the physical frailty and the cognitive impairment was derived. The probability that the study subjects would belong to each subgroup was calculated. To identify factors influencing joint trajectories of physical frailty and cognitive impairment, binomial or multinomial logistic regression was conducted. The GRoLTS checklist in the study (Appendix A).

## 3. Results

### 3.1. Demographic Factors of the Participants

The total number of participants analyzed in this study was 415, with an average age of 72.2 years, and 68.4% of them were aged between 65 and 74 years. Women accounted for 82.9% of the participants (*n* = 344), and most had an education level of elementary school or lower (*n* = 388, 93.5%). Married individuals represented 56.4% of the participants (*n* = 234). Comorbidities were present in 52.3% of the participants (*n* = 217), with 0 or 1 condition. The BMI indicated that 46.5% of the participants were of normal weight, 5.5% were underweight, 22.7% were overweight, and 18.1% were obese. Vision impairment was reported by 47.5% (*n* = 197), while hearing impairment was found in 16.4% (*n* = 68). Subjective health status was poor in 64.3% of the participants (*n* = 267), and the mean depression score was 16.3 (standard deviation = 4.4). Finally, 63.6% of the participants engaged in social activities (*n* = 264).

### 3.2. Number of Trajectory Groups for Physical Frailty and Cognitive Impairment

In the group-based trajectory model, the number of trajectory groups with similar patterns of physical frailty and cognitive impairment was determined by increasing the number of trajectories, one by one, and examining how the goodness-of-fit indices and the quality of classification changed (Table 1). Smaller absolute values of BIC, Sa BIC, and AIC indicate simpler and more explanatory models [38]. When the number of physical frailty trajectory groups increased from three to four, the increase in BIC slowed down. Entropy, which represents the quality of classification, had the closest value to 1, at 0.765, when the number of physical frailty trajectory groups was three, meaning that there were fewer errors in group classification [39]. Therefore, it was concluded that the number of physical frailty trajectory groups was best at three. When examining the probability that participants would belong to each subgroup of the physical frailty trajectory, it was found that 17.9% belonged to the improving physical frailty group, 68.2% belonged to the mild physical frailty group, and 13.9% belonged to the moderate physical frailty group.

Next, when the number of cognitive impairment trajectories increased from two to three, the increase in BIC slowed down, and Entropy had the closest value to 1, at 0.886, when the number of groups was two. Based on this, it was concluded that the number of cognitive impairment trajectory groups was best at two. The probability that participants would belong to each subgroup of the cognitive impairment trajectory was 83.0% for the stable cognitive impairment group and 17.0% for the improving cognitive impairment group.

### 3.3. Joint Trajectory Model of Physical Frailty and Cognitive Impairment

A joint trajectory model of physical frailty and cognitive impairment was estimated in the group-based model, and the parameter estimates for this model are as follows (Table 2). For the physical frailty trajectory model, physical frailty group 1 showed physical frailty at baseline but tended to improve during the follow-up period, indicated by a linear parameter (*p* < 0.001), and was named the improving physical frailty group (IPF). Physical frailty group 2 and group 3 were confirmed to have physical frailty at baseline and continued to have mild or moderate physical frailty during the follow-up period, indicated by a cubic parameter (*p* = 0.160; *p* = 0.419), and were defined as the mild physical frailty group (MPF) and the moderate physical frailty group (MOPF), respectively.

For the cognitive impairment trajectory model, cognitive impairment group 1 was confirmed to have cognitive impairment at baseline and showed a tendency to continue cognitive impairment during the follow-up period, indicated by a cubic parameter (*p* < 0.001), and was named the stable cognitive impairment group (SCI). Cognitive impairment group 2 was confirmed to have cognitive impairment at baseline but showed a tendency to improve during the follow-up period. This was indicated by the identification of three trajectories of physical frailty were identified: the MPF, MOPF, and IPF, and two trajectories of cognitive impairment were identified: the SCI and ICD.

Upon examination of the probability that participants would belong to each subgroup of the physical frailty trajectory, it was found that 16.0% belonged to the IPF, 64.7% belonged to the MPF, and 19.3% belonged to the MOPF. In the figure, dots represent observed data values, and solid lines represent predicted curves. The IPF showed a declining linear parameter (*p* < 0.001) and was defined as the improving physical frailty, indicating an improvement in physical frailty during the follow-up period despite initial physical frailty at baseline. The MPF showed a curve where physical frailty slightly increased at the third-year survey but then continued to be mild. The MOPF showed a curve, which meant maintaining moderate physical frailty without any changes (Figure 2a). Also, the probability that participants would belong to each subgroup of the cognitive impairment trajectory was 83.0% for the SCI and 17.0% for the ICI. During the follow-up period, the SCI showed a curve where cognitive impairment slightly decreased at the second-year survey but then increased again. The ICI showed a declining straight line, indicating improvement in cognitive impairment over time despite initial cognitive impairment at baseline (Figure 2b).

The final joint trajectory model had BIC, Sa BIC, and AIC values of −4885.67, −4856.99, and −4812.68, respectively, and an Entropy value of 0.720, which is close to 1. The final joint trajectory model was not only based on goodness-of-fit indices obtained through statistical analysis but also on the integrated frailty model [22] and clinical practice, and the final decision was made after consulting with a clinical expert group [24].

### 3.4. Demographic, Physical, Psychological, and Social Factors of the Participants According to Physical Frailty and Cognitive Impairment Trajectories

At baseline, the characteristics of the participants according to the trajectory groups of physical frailty and cognitive impairment are as follows (Table 3). The number of comorbidities, visual impairment, subjective health status, depression, and social engagement showed statistically significant differences according to the physical frailty trajectories. The number of comorbidities with two or more was higher in the MOPF, and there was a difference between the groups (*p* < 0.05). Visual impairment was higher in the MOPF at 66.2%. The proportion of subjective healthy status was higher in the IPF, while the proportion of subjective unhealthy status was higher in the MOPF, indicating a difference between the groups (*p* < 0.05). Depression was higher in the MOPF with an average of 18.6 points, and there was a significant difference between the groups (*p* < 0.05). Finally, the proportion of social engagement was highest in the IPF at 66.7%, and there was a significant difference between the groups (*p* < 0.05).

According to the trajectory groups of cognitive impairment, age, sex, education level, marital status, and subjective health status showed statistically significant differences. The proportion of 65–74 year-olds was higher in the ICI, and the proportion of those over 74 years old was higher in the SCI, indicating a difference between the groups (*p* < 0.05). The SCI had a higher proportion of women and those with elementary school education or less compared to the ICI. Married individuals were more prevalent in the ICI. In terms of subjective health status, the ICI had a higher proportion, and there was a difference between the groups (*p* < 0.01).

### 3.5. Factors Associated with Joint Trajectories of Physical Frailty and Cognitive Impairment

Factors associated with physical frailty trajectories were identified, and the results are as follows (Table 4). The IPF was selected as the reference group because it showed improvement in physical frailty over time despite meeting the criteria for physical frailty at baseline, making it appropriate to represent the characteristics of the other two groups. Compared to the IPF, the MPF and MOPF were significantly associated with polypharmacy, overweightness, obesity, and depression. Polypharmacy was 18.49 and 17.66 times (95% confidence interval [CI] 1.10–310.28, *p* < 0.05 and 95% CI 1.07–290.11, *p* < 0.05, respectively) higher than not taking multiple medications. Overweightness was 0.19 and 0.16 times (95% CI 0.05–0.65, *p* < 0.01 and 95% CI 0.05–0.57, *p* < 0.01, respectively) lower, and obesity was 0.22 and 0.18 times (95% CI 0.05–0.86, *p* < 0.05 and 95% CI 0.04–0.70, *p* < 0.05, respectively) lower than being normal weight. Depression was 1.31 and 1.33 times (95% CI 1.13–1.53, *p* < 0.05 and 95% CI 1.14–1.56, *p* < 0.001, respectively) higher. 

Factors associated with cognitive impairment trajectories were identified, and the results are as follows. Compared to those with less than an elementary school education, those with at least a middle school education were 3.32 times (95% CI 1.23–8.95, *p* < 0.05) more likely to be in the SCI. Education level was identified as a factor distinguishing the SCI from the ICI.

## 4. Discussion

Based on the longitudinal data, distinct joint trajectories of physical frailty and cognitive impairment in community-dwelling older adults were identified and their determinants were analyzed. As a result, three trajectories of physical frailty were identified: the MPF, MOPF, and IPF, and two trajectories of cognitive impairment were identified: the SCI and ICD. Factors affecting physical frailty trajectories were found to be the number of medications taken, being overweight, obesity, and depression, while the factor affecting cognitive impairment trajectories was found to be education level. This provided information on the dynamics of physical frailty and cognitive impairment and possible modifying factors.

The findings of this study will contribute to enhancing our understanding of cognitive frailty in Asian older adults and providing a comprehensive knowledge base about its trajectories, thereby increasing the insight of healthcare providers who play a crucial role in community care.

The similar changes in physical frailty and cognitive impairment over time and their common impact on health outcomes support the significance of cognitive frailty. The results suggest that physical frailty and cognitive impairment, which are related to amyloid plaques associated with Alzheimer’s disease, cardiovascular changes, nutritional imbalances including vitamin D, chronic inflammation, and insulin resistance [40,41], may share common mechanisms. The joint trajectory model of physical frailty and cognitive impairment supports the existence of a common underlying mechanism connecting deteriorating physical frailty and cognitive impairment over time [5,41]. Unlike the Fried criteria, which are based on the biological model of frailty proposed by Fried et al. [4], the Frailty Index [6] and the Groningen Frailty Indicator [42] include not only physical components of frailty but also cognitive components. This highlights the need for a standardized measurement tool to assess cognitive frailty, which consists of both physical frailty and cognitive impairment. Currently, there is considerable variation in the methods used to measure cognitive frailty across individual studies, making it difficult to compare and evaluate systematic research findings.

Among the joint trajectory model groups, the study identified the unique groups of IPF and ICI, which were not identified in previous trajectory studies. These unique groups are likely due to differences in the selection criteria of the participants and the characteristics of the study population in previous studies [20,21,43]. One of the selection criteria for the participants in this study was inclusion of those who met the criteria for cognitive frailty at baseline. Additionally, since cognitive frailty is a reversible state before progressing to dementia, which is an early stage of cognitive impairment [10], these unique groups may represent a subgroup showing the reversible nature of cognitive frailty. To confirm the results of this group, additional quantitative and qualitative research is needed to understand the characteristics of the unique groups IPF and ICI that were identified.

Polypharmacy is the greatest risk factor for differentiating the joint trajectory model groups. This is an expected result because the number of medications taken [17] has been consistently reported as a consistent finding in previous studies, and it has been reported in literature reviews [44]. In Korea, the prescription rate of multiple drugs for older adults aged 65 and over has been reported to be high, ranging from 44% to 86% [43,45]. The number of medications taken by older adults in the community is significant because it increases with the aging population and with increases in comorbidities [46]. Polypharmacy is not only likely to lead to an increase in physical frailty but also increases the risk of falls, fractures, dizziness, drug adverse reactions, hospitalization, and mortality [46]. Experts have also emphasized the importance of managing polypharmacy and recommended using standard guidelines (Screening Tool of Older Persons’ Prescriptions [STOPP] or Beers Criteria) centered around doctors to reduce the number of prescribed medications [47]. However, polypharmacy management in clinical settings is still not being addressed. It is time to establish a system for managing polypharmacy in older adults, including the subject, timing, and standardized protocols.

Overweightness and obesity were identified as factors affecting physical frailty trajectories, and compared to normal weight, overweightness and obesity acted as protective factors against physical frailty. Previous studies have reported that abdominal obesity specifically increases the risk of physical frailty, suggesting that it reflects muscle loss, which plays a crucial role in physical frailty [42]. It is necessary to investigate, in more detail, the relationship between body composition and physical frailty by precisely measuring muscle mass, fat mass, visceral fat mass, etc., rather than simply weight or BMI. Moreover, it is necessary to understand the impact of inflammatory markers and hormone levels related to obesity on physical frailty. Through such additional research, a deeper understanding of the complex relationship between obesity and physical frailty can be achieved, ultimately provide information to improve obesity and physical frailty in older adults.

Depression was identified as an important psychological factor in the interaction with physical frailty trajectories. According to systematic literature reviews, older adults in a frail state have a higher risk of depression than those in a non-frail state, and individuals with depression also have a similarly increased risk of physical frailty [9]. This suggests that depression and physical frailty form a complex relationship in which they amplify each other’s risk of occurrence. This finding confirms that chronic physical and psychological stress could be one of the main causes of physical frailty. Furthermore, the need for multifaceted interventions, including the management of depression for improving physical frailty, is emphasized. The interaction between depression and physical frailty should be scientifically verified through randomized experimental studies on the effectiveness of cognitive behavioral therapy or problem-solving therapy recommended [47] in international practice guidelines.

The probability of belonging to the SCI compared to the ICI was significantly influenced by having completed at least middle school education. This result suggests that education level may play an important role in the SCI. Education provides various skills and knowledge that help maintain and improve cognitive function [48]. Skills acquired through education, such as problem-solving abilities, reasoning abilities, and memory, can positively affect the prevention and management of cognitive impairment. However, education alone cannot fully explain the trajectory of cognitive impairment [49]. Therefore, additional research considering other factors, such as the presence of APOE ε4 alleles, income level or occupational status, the presence and management of chronic diseases such as hypertension, diabetes, and dyslipidemia, is needed to fully understand the trajectory of cognitive impairment.

### Limitations, Strengths, and Future Research

This study comprised a secondary data analysis conducted using the KLOSA dataset. The study focused on three out of the five Fried phenotype variables—reduced grip strength, severe fatigue, and unintentional weight loss—to define physical frailty. This approach might limit the generalizability of our findings concerning the physical frailty trajectory. Furthermore, during the screening process, 2281 participants (37.5% of the 6090 eligible individuals aged 65 and older) were excluded due to missing or rejected variables at the baseline assessment, which could potentially introduce selection bias into the study. Despite these limitations, the strength of this study lies in its use of longitudinal data to provide new insights into the trajectories of physical frailty and cognitive impairment, and in identifying polypharmacy, overweightness, obesity, and depression as determinants of physical frailty, and education level as an important factor influencing cognitive impairment. Future research should aim to investigate the long-term effects of interventions targeting these risk factors on the progression of physical frailty and cognitive impairment.

## 5. Conclusions

This study provides evidence for the distinct identification of joint trajectories of physical frailty and cognitive impairment, which can inform the target groups for intervention. The study underscores the role of polypharmacy, overweightness, obesity, and depression as key determinants of physical frailty trajectories, while education level emerged as a significant factor influencing cognitive impairment trajectories. It offers a basis for including modifiable physical and mental factors in intervention components for physical frailty trajectories. Overall, this study advances our understanding of cognitive frailty and its implications for public health strategies aimed at promoting healthy aging.

## Figures and Tables

**Figure 1 healthcare-13-00315-f001:**
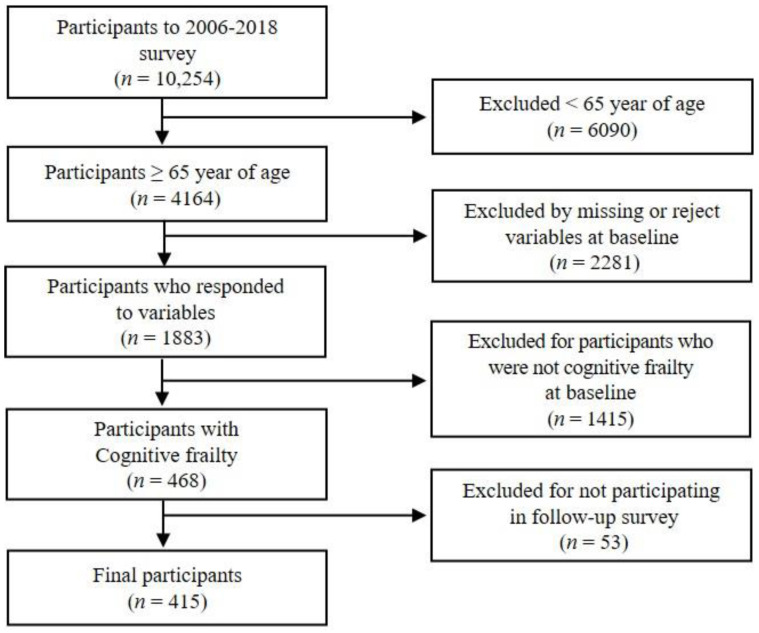
Flow of the selection process for this study sample.

**Figure 2 healthcare-13-00315-f002:**
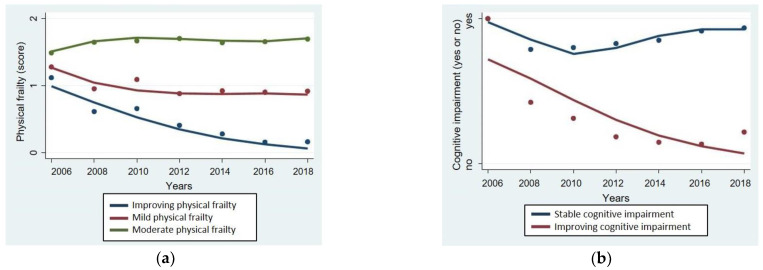
Plots of predicted counts and observed values. (**a**) Physical frailty trajectory groups; (**b**) cognitive impairment trajectory groups.

**Table 1 healthcare-13-00315-t001:** Fit index and classification rate by model according to the number of trajectories of physical frailty and cognitive impairment.

Criteria		Number of Physical Frailty Trajectories				Number of Cognitive Impairment Trajectories			
		2	3	4	5	2	3	4	5
Fit index	BIC	−3678.44	−3653.62	−3651.39	−3666.48	−1238.91	−1259.34	−1182.63	−1272.12
	SaBIC	−3672.64	−3639.12	−3638.82	−3651.01	−1232.28	−1249.87	−1168.43	−1258.87
	AIC	−3660.55	−3608.91	−3612.64	−3618.79	−1218.18	−1229.73	−1138.22	−1230.67
	Entropy	0.608	0.765	0.739	0.537	0.886	0.826	0.623	0.539
Classification rate (%)	1	42.2	17.9	13.2	9.8	83.0	30.8	5.2	3.5
	2	57.8	68.2	19.3	7.9	17.0	63.3	23.2	29.9
	3		13.9	64.8	29.0		5.9	66.7	62.0
	4			2.7	42.8			4.9	2.3
	5				10.5				2.3

AIC: Akaike information criterion; BIC: Bayesian information criteria; SaBIC: Sample-size adjusted Bayesian information criteria.

**Table 2 healthcare-13-00315-t002:** Parameter estimates, classification rate, and fit index for joint trajectory of physical frailty and cognitive impairment.

Group of PF	Parameter	Estimate	SE	*p*	Group of CI	Parameter	Estimate	SE	*p*
1	Intercept	1.25	0.14	<0.001	1	Intercept	7.56	0.82	<0.001
	Linear	−0.31	0.06	<0.001		Linear	−4.87	0.71	<0.001
2	Intercept	1.64	0.16	<0.001		Quadratic	1.14	0.19	<0.001
	Linear	−0.48	0.18	0.006		Cubic	−0.08	0.02	<0.001
	Quadratic	0.09	0.05	0.062	2	Intercept	1.52	0.31	<0.001
	Cubic	−0.01	0.00	0.160		Linear	−0.59	0.11	<0.001
3	Intercept	1.19	0.36	0.001					
	Linear	0.41	0.41	0.320					
	Quadratic	−0.10	0.11	0.374					
	Cubic	0.01	0.01	0.419					
Classification rate (%)		1	16.0	(n = 60)	Classification rate (%)		1	83.0	(n = 345)
		2	64.7	(n = 287)			2	17.0	(n = 70)
		3	19.3	(n = 68)					
Fit index		BIC	−4885.67						
		SaBIC	−4856.99						
		AIC	−4812.68						
		Entropy	0.720						

AIC: Akaike information criterion; BIC: Bayesian information criteria; CI: Cognitive impairment; PF: physical frailty; SaBIC: Sample-size adjusted Bayesian information criteria; SE: Standard error.

**Table 3 healthcare-13-00315-t003:** Characteristics of the participants according to the joint trajectory of physical frailty and cognitive impairment at baseline.

Variable	Physical Frailty						*p* ^b^	Cognitive Impairment				*p* ^b^
	IPF (*n* = 60)		MPF (*n* = 287)		MOPF (*n* = 68)			SCI (*n* = 345)		ICI (*n* = 70)		
Age												*
65–74 years	40	66.7%	201	70.0%	43	63.2%		228	66.1%	56	80.0%	
over 75 years	20	33.3%	86	30.0%	25	36.8%		117	33.9%	14	20.0%	
Sex												*
man	13	21.7%	48	16.7%	10	14.7%		52	15.1%	19	27.1%	
woman	47	78.3%	239	83.3%	58	85.3%		293	84.9%	51	72.9%	
Education level ^a^												**
≤primary school	56	93.3%	268	93.4%	64	94.1%		329	95.4%	59	84.3%	
≥middle school	4	6.7%	19	6.6%	4	5.9%		16	4.6%	11	15.7%	
Marital status												*
married	34	56.7%	164	57.1%	36	52.9%		186	53.9%	48	68.6%	
unmarried	26	43.3%	123	42.9%	32	47.1%		159	46.1%	22	31.4%	
Comorbidity							*					
0 or 1	41	68.3%	145	50.5%	31	45.6%		180	52.2%	37	52.9%	
≤2	19	31.7%	142	49.5%	37	54.4%		165	47.8%	33	47.1%	
Polypharmacy												
0	28	46.7%	117	40.8%	20	29.4%		135	39.1%	30	42.9%	
1–2	31	51.7%	144	50.2%	42	61.8%		182	52.8%	35	50.0%	
≥3	1	1.6%	26	9.0%	6	8.8%		28	8.1%	5	7.1%	
BMI ^a^												
normal weight	22	37.9%	139	53.1%	32	49.2%		157	49.5%	36	52.9%	
underweight	3	5.2%	14	5.3%	6	9.2%		20	6.3%	3	4.4%	
overweight	22	37.9%	60	22.9%	12	18.5%		75	23.7%	19	27.9%	
obesity	11	19.0%	49	18.7%	15	23.1%		65	20.5%	10	14.8%	
Vision impairment ^a^							**					
no	37	61.7%	157	54.9%	23	33.8%		173	50.3%	44	62.9%	
yes	23	38.3%	129	45.1%	45	66.2%		171	49.7%	26	37.1%	
Hearing impairment												
no	53	88.3%	235	81.9%	59	86.8%		284	82.3%	63	90.0%	
yes	7	11.7%	52	18.1%	9	13.2%		61	17.7%	7	10.0%	
Physical activity												
yes	9	15.0%	55	19.2%	11	16.2%		58	16.8%	17	24.3%	
no	51	85.0%	232	80.8%	57	83.8%		287	83.2%	53	75.7%	
Alcohol consumption												
no	50	83.3%	240	83.6%	58	85.3%		291	84.3%	57	81.4%	
yes	10	16.7%	47	16.4%	10	14.7%		54	15.7%	13	18.6%	
Subjective health status							*					**
good	29	48.3%	102	35.5%	17	25.0%		113	32.8%	35	50.0%	
poor	31	51.7%	185	64.5%	51	75.0%		232	67.2%	35	50.0%	
Depression (Mean ± SD)	15.5	3.4	16.0	4.4	18.6	4.5	*	16.5	4.4	15.4	4.6	
Social engagement							*					
yes	40	66.7%	190	66.2%	34	50.0%		218	63.2%	46	65.7%	
no	20	33.3%	97	33.8%	34	50.0%		127	36.8%	24	34.3%	

BMI: Body mass index; ICI: Improving cognitive impairment; IPF: Improving physical frailty; MPF: Mild physical frailty; MOPF: Moderate physical frailty; SCI: Stable cognitive impairment; SD: Standard deviation. ^a^ Analysis included missing data. ^b^
*p*-value for Pearson’s chi-square test. * *p* < 0.05, ** *p* < 0.01.

**Table 4 healthcare-13-00315-t004:** Associated modifiable factors with joint trajectories of physical frailty and cognitive impairment.

Variable	MPF vs. IPF (Ref.)				MOPF vs. IPF (Ref.)				SCI vs. ICI (Ref.)			
	RRR		95% CI		RRR		95% CI		RRR		95% CI	
Age (ref. 65–74 years)	1.00				1.00				1.00			
over 75 years	0.75		0.39	1.45	0.56		0.20	1.58	0.52		0.26	1.04
Sex (ref. man)	1.00				1.00				1.00			
woman	1.55		0.60	3.99	0.53		0.11	2.43	0.67		0.30	1.48
Education level (ref. ≤ primary school)	1.00				1.00				1.00			
≥middle school	0.42		0.04	4.61	0.61		0.06	6.14	3.32	*	1.23	8.95
Marital status (ref. married)	1.00				1.00				1.00			
unmarried	1.40		0.47	4.12	1.28		0.44	3.76	0.75		0.40	1.41
Comorbidity (ref. 0 or 1)	1.00				1.00				1.00			
≤2	1.63		0.55	4.79	1.75		0.60	5.14	1.63		0.80	3.33
Polypharmacy (ref. 0)	1.00				1.00				1.00			
1–2	3.03		0.84	10.93	3.21		0.90	11.38	0.76		0.37	1.55
≥3	18.49	*	1.10	310.28	17.66	*	1.07	290.11	0.70		0.19	2.53
BMI (ref. normal weight)	1.00				1.00				1.00			
underweight	2.18		0.33	14.61	2.31		0.35	15.31	0.61		0.15	2.41
overweight	0.19	**	0.05	0.65	0.16	**	0.05	0.57	1.03		0.53	2.00
obesity	0.22	*	0.05	0.86	0.18	*	0.04	0.70	0.74		0.33	1.69
Vision impairment (ref. no)	1.00				1.00				1.00			
yes	1.71		0.60	4.88	1.63		0.57	4.71	0.85		0.46	1.55
Hearing impairment (ref. no)	1.00				1.00				1.00			
yes	1.59		0.37	6.81	1.78		0.41	7.73	0.53		0.20	1.36
Physical activity (ref. yes)	1.00				1.00				1.00			
no	1.08		0.25	4.58	1.15		0.27	4.89	0.80		0.39	1.61
Alcohol consumption (ref. no)	1.00				1.00				1.00			
yes	1.47		0.34	6.33	1.40		0.33	6.01	0.78		0.33	1.83
Depression	1.31	***	1.13	1.53	1.33	***	1.14	1.56	0.97		0.90	1.05
Subjective health status (ref. good)	1.00				1.00				1.00			
poor	0.83		0.24	2.85	0.88		0.25	3.10	0.73		0.38	1.41
Social engagement (ref. yes)	1.00				1.00				1.00			
no	2.10		0.79	5.59	2.23		0.81	6.10	1.07		0.60	1.93

BMI: Body mass index; CI: Confidence interval; ICI: Improving cognitive impairment; IPF: Improving physical frailty; MPF: Mild physical frailty; MOPF: Moderate physical frailty; RRR: Relative risk ratio; SCI: Stable cognitive impairment. * *p* < 0.05, ** *p* < 0.01, *** *p* < 0.001.

## Data Availability

The data for this study used the KLOSA 2006–2018. The researcher accessed the homepage of the Employment Survey Analysis System (https://survey.keis.or.kr/) on 14 June 2020.

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
