# Peer review of "Impact of Modifiable Factors Associated with Physical Frailty and Cognitive Impairment Trajectory of Older Adults: Using the Korean Longitudinal Study of Aging 2006–2018"

_healthcare, 2025, doi:10.3390/healthcare13030315_

Round 1

Reviewer 1 Report

Comments and Suggestions for Authors

The study aimed to identify joint trajectories of physical frailty and cognitive impairment among community-dwelling older adults and to determine modifiable factors for each trajectory. Here are suggestions for improving the text:

Abstract

1. Enter the average age of the participants.

Introduction

1. Between lines 55-59 the authors seek to justify the performance of this study. The following sentence requires the inclusion of a theoretical framework: "However, previous studies lack comprehensive research on the general health status of older adults, including health-promoting behaviors, psychological factors, and social factors, which can predict cognitive frailty, as well as all the statements in the paragraph". Therefore, the remaining sentences in this paragraph need to be referenced by studies.

2. The following sentence also needs a theoretical basis: "Previous studies (Where are these studies???) have primarily focused on cross-sectional surveys to explore the incidence rate or determinants of cognitive frailty, which are based on short-term analyses that focus on differences in individual averages."

3. In general, this section needs to be better organized in relation to the passages that seek to justify the study;

4. I suggest including studies from your country so that readers have a better idea of ​​what exists in Korea and, mainly, the gaps in knowledge in Korea about frailty and cognition during aging.

Methods

This section is rigorous and well organized.

Results

1. Table 1 should be presented after the introduction of the RESULTS caption and not after "2.5. Statistical Analysis". I suggest placing Table 1 at the end of the paragraph after its citation (line 250)

Discussion

1. I suggest including a new section at the end of the Discussion, titled "Limitations, strengths, and future research";

2. Your study has no limitations, this is a weak point of the presentation;

3. A series of suggestions for future studies should be conducted;

4. Perhaps a strong point of the study is that it is the first in Korea to develop longitudinal analyses on physical and cognitive frailty. If so, please let us know!

Author Response

Reviewer 1

I am glad that you gave me the opportunity to improve my research by giving me comments on my manuscript. The response to your comments is as follows:

Reviewer 2 Report

Comments and Suggestions for Authors

The present study is promising, however some improvements are needed. Throughout the text appears the expression "we"... But the article has only one author...

Please see other comments below…

INTRODUCTION

P1L29. “According to the results of the domestic elderly status survey...” – The domestic elderly status survey from where?

MATERIALS AND METHODS

It is not clear for me why only participants who met the criteria for cognitive frailty at the time of the first-wave survey were included.

P3L136. “We define cognitive frailty when older adults meet the criteria for both physical frailty and cognitive impairment." - Based on whom?

It is not clear, either in the Introduction or in the Methods section, how frailty in the elderly can be assessed...

P3L137. “The Fried phenotype includes five operational definitions: reduced grip strength, severe fatigue, unintentional weight loss, low activity, and slow walking speed. Older adults are defined as frail if they meet three or more of these criteria [4]. We define physical frailty using available variables from the KLOSA, including reduced grip strength, severe fatigue, and unintentional weight loss, and we consider older adults as frail if they meet one or more of these criteria [16]." - There seems to be a lack of coherence between these two references... Thus, it is not clear what the author's idea and decision are...

P3L146. “Second, severe fatigue was assessed using two questions from the Center for Epidemiological Studies Depression Scale..." - This tool to assess severe fatigue is based on what reference?

P3L146. “Third, unintentional weight loss was defined as answering "lost weight" to the question about weight change of 5 kilograms or more in the past year." - Based on whom?

P5L188. “Physical activity was defined as engaging in exercise at least once a week..." - But this is clearly out of step with what the current guidelines for physical activity are... 

RESULTS

Table 1. - It is not clear, either in Methods or Results section, how individuals are classified as improving PF, Mild PF, Moderate PF, Stable CI, Improving CI... Moreover, it's important to explain what this really means...

DISCUSSION

P5L188. “As a result, we identified three trajectories of physical frailty: MPF, MOPF, and IPF, and two trajectories of cognitive impairment: SCI and ICD...." - They should be among the first results to be presented... And this is not clear in the Results section...

Author Response

Reviewer 2

I am glad that you gave me the opportunity to improve my research by giving me comments on my manuscript. The response to your comments is as follows:

Round 2

Reviewer 1 Report

Comments and Suggestions for Authors

Dear all, I consider that the study has been well done.

However, I still ask that the word “elderly” (line 29) be replaced by “older adult”. And I also suggest checking the entire text to make sure that this word is not repeated.

Reviewer 2 Report

Comments and Suggestions for Authors

It is necessary in the Introduction section to introduce a paragraph that approach the assessment of frailty in the elderly, justifying your options in the Methods section...

Considering the limitations of the physical activity variable in the present study, I suggest to eliminate this variable of the study...

To enhance reader comprehension, I suggest to restructuring the presentation of data.

Author Response

Thank you for providing valuable feedback on my manuscript. Your comments have been carefully considered, and I have made the necessary revisions to ensure that the introduction provides sufficient background and includes all relevant references.
